# Dynamic Arches Destruction by a Bulk Material Flow Separator: A Case Study of the Separator Usage in Microwave Grain Processing Plants

**Alexey A. Vasilyev** *[ID], **Alexey N. Vasilyev** [ID], **Dmitry Budnikov** [ID], **Vadim Bolshev** *[ID], **Denis Shilin** [ID] **and Dmitry Shestov**

Federal Scientific Agroengineering Center VIM, 109428 Moscow, Russia; vasilev-viesh@inbox.ru (A.N.V.); dimm13@inbox.ru (D.B.); deninfo@mail.ru (D.S.); shestov.d.a@mail.ru (D.S.)
* Correspondence: lex.of@mail.ru (A.A.V.); vadimbolshev@gmail.com (V.B.); Tel.: +7-499-174-85-95 (A.A.V. & V.B.)

**Abstract:** Hoppers for unloading bulk materials are an indispensable feature of many technological machines and not only those employed in agricultural production. One of the problems in the operation of hoppers is the appearance of dynamic arches which make the outflow of grain uneven. Experimental studies have previously shown that the formation of dynamic arches creates an uneven outflow of grain along the vertical zones of outlet hoppers. This can lead to the processing mode violation of bulk material and loss of machine productivity. This paper theoretically shows that the use of arch-breaking devices with an element oscillating in the grain layer requires systems for automatic adjustment of oscillation frequency. This complicates the design and makes it more expensive. It is proposed to amend the type of material outflow as it moves along the hopper's height to solve this problem. The possibility of employing this approach has been tested on the use of a bulk material flow separator in the outlet hopper. The bulk material flow separator is made in a plate form and is installed rigidly between two opposite walls of the outlet hopper. Thus, the volume of the hopper is divided into two vertical equal parts. With the help of experimental studies, it was determined that the lower side of the bulk material flow separator should be at a distance of 0.3 of the total hopper height from the discharge opening, while the upper side of the separator should be at a distance of 0.25 of the hopper's height from the top edge of the hopper. The experimental verification of flow separator use confirmed its effectiveness.

**Keywords:** outlet hopper; microwave–convective processing; dynamic arch; arch destruction; bulk material; grain; grain flow; bulk material separator

## 1. Introduction

In the technological processes of agriculture, storage hoppers are used in a wide range of equipment. For example, in grain cleaning machines the grain is located in a storage hopper before it enters the cleaning system [1]. The dispensers also contain hoppers where a product is stored before the distribution of the required portion [2,3]. In sorting, grain is stored in the hopper before reaching the sorting equipment [4,5]. When sowing seeds, it is necessary to have a sufficient supply of grain and fertilizers in a seeder hopper to ensure operation on the required area [6,7]. The hopper is one of the mandatory elements of technological equipment in most agricultural machines. In addition, when operating these assemblages of equipment, it is necessary to ensure a uniform supply of bulk material from the hopper to the working bodies. Without this condition being met, the productivity of machines decreases (conveyors, grain cleaners, sorting equipment) and the accuracy and work quality of technological equipment (grain cleaners, seeders, dispensers) are compromised. However, with technological equipment, there is necessarily a problem of ensuring the uniformity of the outflow of bulk materials from the storage hopper [8,9]. The uneven flow of bulk material from outlet hoppers occurs due to the formation of

dynamic arches in them [10,11]. In turn, the formation of arches is due to the compaction of bulk products when they are loaded into the hopper and depends on seed culture, granulometric composition and humidity [12]. The arch is a kind of dome made of loose material particles. It withstands the pressure of the bulk material located on the top and prevents its movement. At the same time, bulk material located below the arch moves freely to the outlet opening of the hopper. After some time, the dynamic arch collapses and the bulk material continues to move down.

The outflow process of bulk materials from the outlet hoppers of technological machines has been studied in sufficient detail [13,14]. In our case, we consider the stationary outflow process which is represented by two outflow types—normal and hydraulic [15]. The combination of these types is called a mixed type of bulk materials outflow. When the normal outflow of bulk material is carried out, a channel is formed above the outlet opening where the grain moves from the hopper walls. When a hydraulic outflow occurs, the entire volume of bulk material moves evenly over the surface bulk material area to an outlet. In this paper, technological plants are considered for microwave–convective grain processing. It is crucial for them that grain movement in the microwave–convective zone occurs evenly. Therefore, for such plants the hopper design must provide the hydraulic type of grain outflow. The formation frequency of dynamic arches depends on the product type, its moisture content and the outlet hopper parameters. Therefore, there are a number of works that consider solutions to the problem of a uniform outflow of grain from the hopper outlets, including dynamic arch formation [16,17]. In our previous studies [12,18], we have shown that the dynamic arch formation in outlet hoppers of technological equipment not only leads to uneven bulk material outflow but also to uneven movement in equipment located above an outlet hopper. For some technological processes, this is even more of a problem than uneven grain flow.

## 2. Materials and Methods

Microwave–convective plants for bulk material processing are installed above an outlet hopper, as shown in Figure 1. In such plants, a serious processing violation mode of technological material may occur while working with microwaves [19].

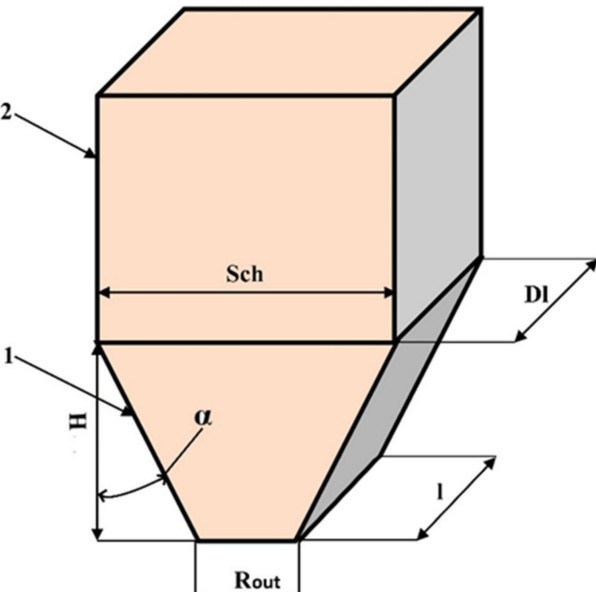

**Figure 1.** Schematic design of technological equipment for grain processing with an outlet hopper: (1) outlet hopper; (2) microwave–convective module for processing bulk materials; (H) outlet hopper height; (Sch) width of upper front face of hopper and lower face of microwave–convective module; (Dl) length of lateral face of hopper and lower lateral face of microwave–convective module; (Rout) hopper outlet width; (l) hopper outlet length; (α) inclination angle of outlet hopper wall.

Figure 1 depicts outlet hopper 1 of the microwave–convective module 2. Outlet hopper height H, inclination angle of outlet hopper wall $\alpha$ and hopper outlet width Rout are chosen so as to ensure a uniform grain outflow [9,20,21]. However, such a hopper construction does not exclude the formation of a dynamic arch in an outlet hopper, leading to uneven movement of grain in the microwave–convective zone. In this case, one volume of grain is exposed to the microwave field for a longer time than the other. The recommended modes of grain processing are violated and this leads to a decrease in technological equipment productivity and to grain quality deterioration. This situation is dangerous both for drying grain and for disinfection and pre-sowing treatment of seeds. To prevent such a situation, it is necessary to deal with the formation of dynamic arches.

Arch-breaking devices can be used to counteract the formation of dynamic arches [22,23]. These devices consist of several elements wherein one of them performs various types of movement in the grain layer located in a hopper. The resulting dynamic arches are destroyed due to these movements. The element movement in the grain layer is usually carried out by an electric drive. The use of such arch-breaking devices [24] requires additional serious structures and control systems. The paper investigates the possibility of ensuring dynamic arch destruction in outlet hoppers without involving additional equipment.

The experimental model of an outlet hopper was made in such a way as to be able to visually observe and record grain movement along the hopper's height during its unloading. To achieve this, the experimental model (layout) was made of Plexiglass. The side walls of the hopper layout were divided into squares $2 \times 2$ cm in size so that it was possible to accurately determine the coordinates of grain movement.

The experimental model was a part (section) of the outlet hopper 5 cm thick. The parameters of the model were calculated in accordance with the parameters of technological plants for microwave–convective grain processing (length and width). The productivity of the plant for microwave–convective drying of spiked crops was 15 t/h. The inclination angle of bunker walls was calculated based on the fact that the plant would process the grain of wheat, barley and sunflower [12]. For these crops, the following were taken into account:

— Laying angle was 17° for wheat, 18° for barley, 16° for sunflower;
— Internal friction angle was 16° for wheat, 15° for barley, 19° for sunflower;
— Reduced friction angle was 27° for wheat, 39° for barley, 26° for sunflower.

As a result of calculations, the width of outlet opening adopted was equal to 90 mm, while the height of the hopper was 690 mm. A cut was made along the vertical central axis of the hopper, which made it possible to build a plate inside so as to divide the layout into two equal parts. The total volume of the bunker was 0.0082 m³.

The transparent walls of the experimental model allowed filming the movement of grain along the hopper's height. The video played in slow motion in order to record the change in the height of the moved grain layer. Subsequently, the obtained data were presented in the form of graphs.

## 3. Theoretical Research

The schemes of bulk material movement in outlet hoppers along slip lines parallel (equidistant) to the generatrix of the flow sliding surface are adopted in the developed theory of dynamic arch formation and destruction (Figure 2) [14,25].

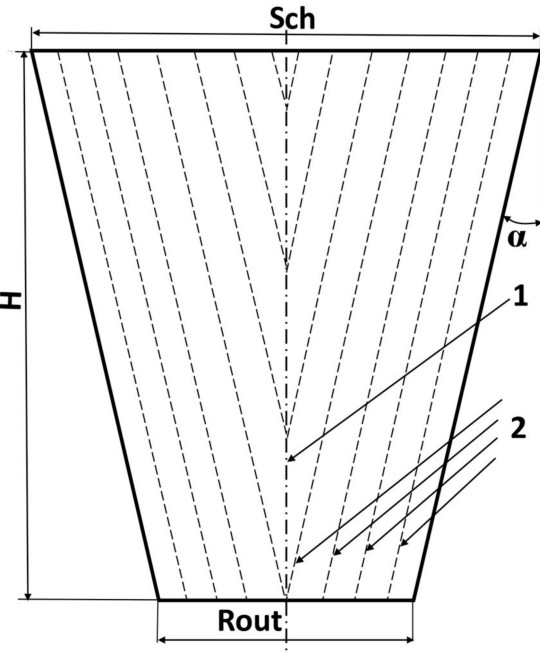

**Figure 2.** Scheme of bulk material movement in an outlet hopper: (1) hopper vertical centerline; (2) lines equidistant from the generatrix of the flow sliding surface.

Dynamic arches are formed in the bunker due to such a scheme of bulk materials movement. The frequency of their formation depends on several parameters. One of these parameters is the arching coefficient $K_{arch}$ [24]:

$$K_{arch} = \frac{\lambda_0 (V_d + V_e)}{\lambda_0 V_d + (1 + \lambda_0) V_e},$$

(1)

where $V_e$ is bulk material volume in the under-arch space "e", which is equivalent to an unstable arch (Figure 3), m$^3$; $V_d$ is bulk material volume in the under-arch space "d" (Figure 3), m$^3$; and $\lambda_0$ is a coefficient characterizing the relative axial compliance of an unstable arch.

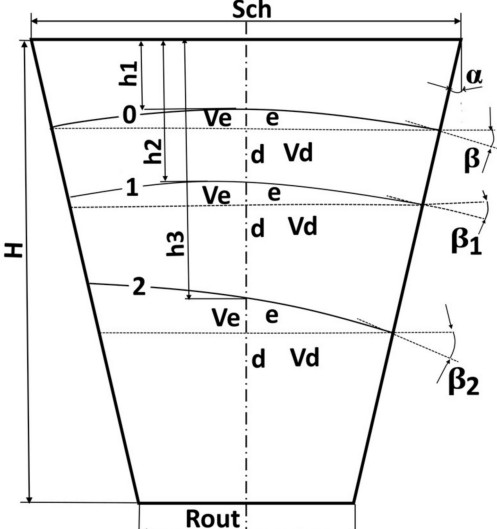

**Figure 3.** An example of dynamic arch formation in an outlet hopper when moving bulk material.

The coefficient $\lambda_0$ can be found by Equation (2):

$$\lambda_0 = \frac{tg\alpha - tg\beta}{tg\alpha},\tag{2}$$

where $\beta$ is the angle between the tangent to the arch curve and the horizontal at any height h of the bulk material flow (Figure 3).

The angle $\beta$ can be calculated by Equation (3):

$$\beta = arctg\frac{H\sin^2\alpha}{0.5Sch(1 + tg^2\alpha) - Htg^3\alpha},\tag{3}$$

Equation (1) was constructed for the hydraulic material outflow, assuming that the resulting dynamic arches do not vary their angle of inclination as they move along the height of the outlet hopper (arch "0" in Figure 3). In this case, the angle $\beta$ remains constant over the entire hopper's height, H. However, dynamic arches were discovered to modify the angle of their inclination to the horizontal when moving from top to bottom, both theoretically [26] and empirically [18] (arches "1" and "2" in Figure 3). Research results show there is actually the dependence:

$$\beta = f(h),\ V_e = f(h),\ V_d = f(h),$$

where h is the distance from the top edge of the outlet hopper to the dynamic arch (arch formation depth).

As a consequence, $\lambda_0 = f(h)$, $K_{arch} = f(h)$. Only the angle $\alpha$ remains constant, which determines the angle of bunker wall inclination and, accordingly, the lines equidistant to the generatrix of the flow sliding surface along which the grain moves in during unloading a hopper. To take into account these features, it is necessary to make significant adjustments to the elements of the theory of calculating dynamic arches and determining methods for counteracting their formation. Some research [27,28] has aimed at the development of arch-breaking devices. As a rule, a feature of these devices is the movement of the arch-destroying element with the frequency of arch formation. Thus, this frequency is necessary to know. Equation (4) can be used to determine this frequency thorough calculating the lifetime of an unstable arch. The time during which an unstable arch is located at height $h$ can be equal to the time during which bulk material with a total volume $V_e$ and $V_d$ flows out of a hopper from an under-arch space [25]:

$$T_p = \frac{\lambda_0 V_d + (1 + \lambda_0)V_e}{\lambda_0 q} = \frac{V_d + \frac{V_e}{\lambda_0} + V_e}{q},\tag{4}$$

where $T_p$ is pulsation time, s; q is the flow rate, that is, bulk material outflow from an outlet hopper, $m^3/s$.

A hopper with a slotted outlet is used in microwave–convective grain processing plants. Such a design for the hopper makes it possible to ensure the hydraulic outflow of the material and control the grain flow. The differential Equation (5) describes the change in the flow rate of bulk material from the slotted hopper outlet [24]:

$$K_{arch}\frac{dq}{d\tau} + \frac{tg\alpha}{2glR_{out}^2}K_{arch}^2q^2 = 2lR_{out},\tag{5}$$

where $R_{out}$ is the width of the hopper outlet, m; l is length of hopper outlet, m; g is free fall acceleration, $m/s^2$; $\tau$ is time, s.

Equation (5) shows that the hopper is a non-linear object. After solving Equation (5), the following equation was obtained:

$$q = K_{arch} 2l \frac{th\left(\sqrt{\frac{tg\alpha}{R_{out}g}}\right)}{\sqrt{tg\alpha}} \cdot \tau \sqrt{\frac{R_{out}^3 g}{tg\alpha}}, \tag{6}$$

Equation (6) describes the change in bulk material flow from a hopper outlet in a steady state. In this case, the regime is considered to be steady when the flow rate remains constant and there is no change in the position of the outlet valve. However, in the process of operation of a microwave–convective plant for grain processing, the consumption of bulk material is one of the regulatory influences on changes in grain moisture. Therefore, the outlet valve must periodically change its position during the plant operation. The system of Equation (7) mathematically describes such change in grain flow.

$$q = \begin{cases} q_{st} - e^{-\tau T_p}, \text{ when } \tau \leq \tau_{st} \\ q = K_{arch} 2l \frac{th\left(\sqrt{\frac{tg\alpha}{R_{out}g}}\right)}{\sqrt{tg\alpha}} \cdot \tau \sqrt{\frac{R_{out}^3 g}{tg\alpha}}, \text{ when } \tau > \tau_{st} \end{cases}, \tag{7}$$

where $q_{st}$ is the steady-state value of the bulk material flow rate after the completion of the transition process, m$^3$/s; and $\tau_{st}$ is the time for transient process completion, s.

The system of Equation (7) reflects the change in the flow rate of bulk material at the outlet of the hopper in two stages. The first stage represents the transient process, when the outlet valve has changed its position. As a result, the flow rate of bulk material changes to the steady-state value of the bulk material flow rate after the completion of the transition process $q_{st}$. The second stage is the grain outflow from the outlet opening, when the valve remains stationary while the flow rate changes only due to the formation of dynamic arches in a hopper.

An analysis of Equations (1)–(6) shows that most of parameters do not remain constant in the process of moving bulk material along the hopper's height. Therefore, there cannot be a constant oscillation frequency of an arch-breaking device inside the grain layer. The effect of the operation of an arch-breaking device will be local. This device will effectively destroy dynamic arches if they are formed with the frequency of device operation. To ensure the effective operation of such an arch-breaking device, it is necessary to use an automatic system for changing the frequency of its oscillations. In turn, this will require the development of sensors for the occurrence of dynamic arches. In this case, the system complexity of the arch-breaking device increases many times over. In [29], automated systems for controlling the frequency of arch-breaking devices were developed, which can change the oscillation frequency only for a specific type of grain (wheat, corn, millet, etc.). At the same time, it is difficult to assume what will be the uniformity of grain movement along the vertical halves of the outlet hopper. No such studies have been conducted. Therefore, the use of arch-breaking devices cannot be considered an effective means to ensure uniform grain movement in an outlet hopper. In the current paper, a solution to this problem was attempted using a different approach.

Based on the material submitted, a hypothesis was put forward that changing the angle $\alpha$ in the process of moving grain along the hopper's height will allow control of the angle of dynamic arch inclination and ensure the movement of equal grain volumes along the right and left vertical halves of the outlet hopper. This hypothesis can be implemented by changing the angle of the hopper wall inclination $\alpha$ along with its height. However, this decision will complicate the hopper design. The same effect can be achieved by changing the type of bulk material flow in a hopper. As previously noted, the size of the outlet $R_{out}$ and the angle $\alpha$ are taken to ensure the hydraulic material outflow from a hopper. Therefore, it is necessary to add an element to the design of the hopper that would change the type of material outflow. A bulk material flow separator can be used as such an element, made in a plate form rigidly installed between the hopper walls (Figure 4).

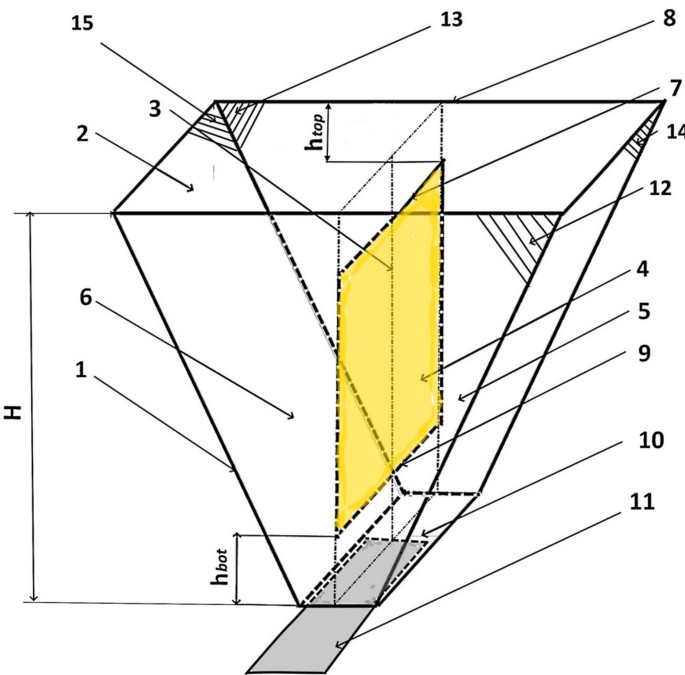

**Figure 4.** Schematic design of the hopper with a bulk material flow separator: (1) hopper; (2) loading window; (3) central vertical axis; (4) bulk material flow separator; (5) and (6) vertical hopper parts equal in volume; (7) upper side of bulk material flow separator; (8) top of hopper wall; (9) bottom side of bulk material flow separator; (10) outlet window; (11) valve; (12) and (13), (14) and (15) opposite side walls of hopper.

The bulk material flow separator 4 is made in the plate form rigidly attached to the opposite walls (12) and (13) of a hopper in such a way that it divides its volume into two equal vertical parts (5 and 6). The upper side (7) of the bulk material flow separator (4) is located at a distance $h_{top}$ of height H from the top of the hopper's wall (8). The lower side (9) of the bulk material flow separator (4) is located from the outlet window of the hopper (10) at a distance $h_{bot}$ of hopper height H.

Such fixing of the bulk material flow separator will allow change of the grain type outflow in a hopper. Figure 5 shows a diagram of the grain movement in a hopper with the bulk material flow separator.

In the upper part of the outlet hopper (before the bulk material flow separator), grain is moved along lines equidistant to the generatrix of the hopper wall surface (position 2 in Figure 5) at an angle of inclination $\alpha$. The grain movement in this zone is carried out as it is in a hydraulic material outflow. When grain enters the material flow separator area, the type of grain outflow changes to normal. In this case, the grain movement trajectories change as it approaches the bulk material flow separator. Thus, the closer to the hopper wall, the closer the trajectory angle of grain movement to the angle $\alpha$. When moving closer to the hopper center, the grain movement trajectories 3 and 4 (Figure 5) will change the angles of inclination from $\alpha_2$ to $\alpha_{2'}$ in the upper part of the zone, and from $\alpha_{2''}$ to $\alpha_{2'''}$ in the lower part of the zone. When the grain moves below the bulk material flow separator, the trajectory of the material movement comes close to the angle $\alpha$ (position 2 in Figure 5). Therefore, a hydraulic type of grain outflow is ensured. The option of using a bulk material flow separator with grain outflow type changes can ensure the movement of the same grain volumes in the outlet hopper vertical parts 5 and 6 (Figure 4).

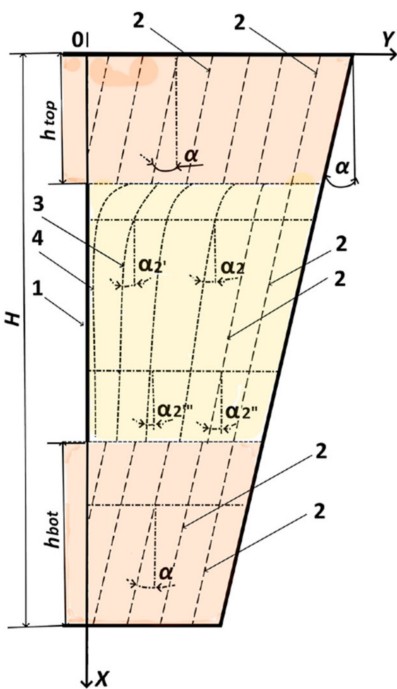

**Figure 5.** Scheme of changing a bulk material outflow type in the presence of a bulk material flow separator in a hopper: (1) bulk material flow separator; (2) lines equidistant from the generatrix of the flow sliding surface; (3) central vertical axis; (4) trajectories of the bulk material flow near the separator.

## 4. Experimental Research

Experimental studies were carried out in order to determine the values of $h_{top}$ and $h_{bot}$, which ensure uniform grain outflow from each vertical half of the outlet hopper. An experimental setup in the form of a hopper layout made of Plexiglas were used for the study. The walls of the setup are marked with lines at 2 cm intervals [18]. A vertical slit was made in the hopper wall's layout along its central axis, into which a bulk material flow separator was inserted. The length of the bulk material flow separator was changed in the course of the experiment so as to obtain a variety of results with a combination of $h_{top}$ and $h_{bot}$ values.

The experiment was carried out in two stages. In the first stage, it was determined what the distance should be from the outlet to the lower side of the bulk material flow separator, $h_{bot}$, in order to ensure the least uneven volumes of grain moving along the vertical parts 5 and 6 (Figure 4) of a hopper.

The distance was determined in relative units:

$$h_{top} = h_{top}(cm)/H(cm), \tag{8}$$

$$h_{bot} = h_{bot}(cm)/H(cm), \tag{9}$$

For this, $h_{top} = 0$ was taken, while $h_{bot}$ was changed in the range 0–0.35 H with an interval of 0.07 H.

The experiment was carried out in the following sequence: a material flow separator was installed in the hopper layout; the outlet valve was closed; grain was poured into the bunker layout; the outlet valve was opened; the process of moving grain in the bunker layout was filmed.

A data set was obtained reflecting the dependence dh $= f(h)$ during video processing. Where dh is the difference in distances h from the grain surface in the vertical hopper parts to the top of the hopper wall (Figure 6).

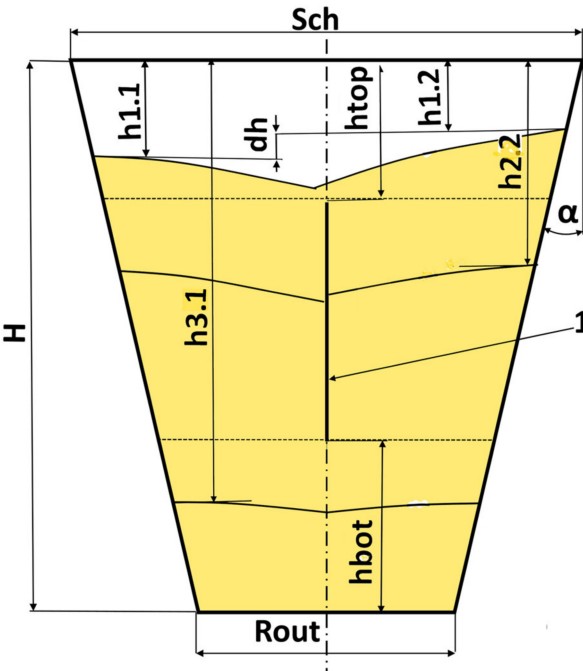

**Figure 6.** Controlled parameters on a hopper layout during the experiment: (1) bulk material flow separator.

For example, Figure 6 shows the difference in distances dh from the grain surface to the top of the hopper wall in its left part h1.1 and the distance from the grain surface to the top of the hopper wall in its right part h1.2, that is, dh = h1.1 − h1.2. This value is dimensional and expressed in cm. Figure 6 also shows the distance to the grain surface when the grain is in the area where the bulk material flow separator is installed h2.2 (on the right side of the hooper) and when the grain is below the area with the bulk material flow separator h3.1 (on the left side of the hopper). The number and location of the grain surfaces in Figure 6 are chosen arbitrarily to demonstrate the measurement technique during the experiment.

As a result of the experimental research, it was found that changing the distance $h_{bot}$ significantly affects the uneven movement of grain in an outlet hopper. The least unevenness of grain movement along the vertical hopper parts is observed for $h_{bot} = 0.3$ H. Therefore, this ratio is accepted as recommended.

At the next stage of the experimental research, it was necessary to determine what should be the distance $h_{top}$ from the top of the hopper wall to the top side of the bulk flow separator. In this case, the distance $h_{bot}$ was fixed at a distance of 0.3 H. The experiment was carried out with a varying value of $h_{top}$ in the range 0–0.28 H (0, 0.09; 0.18; 0.22; 0.26; 0.28). This combination was adopted because a wider range of distance combinations fell into the study area.

The experimental methodology was the same as when determining the recommended $h_{bot}$ distance. The distance from the grain surface to the upper edge of the hopper, $h_{rel}$, was chosen for the left vertical hopper half expressed relative to the total height of the outlet hopper, H. This distance was taken as a reference to simplify the measurement process and make it clear. So, in Figure 6 for the first grain surface, the required distance was determined as:

$$h_{rel} = h1.1/H, \tag{10}$$

While moving grain along the outlet hopper height, the distance $h_{rel.i}$ was:

$$h_{rel.i} = h1.i/H, \tag{11}$$

where i is the number of the grain surface as grain moves along the outlet hopper height; and $h_{rel.i}$ is the relative distance of the ith grain surface to the upper edge of the outlet hopper.

Data were obtained as a result of processing the video material and are displayed in the form of graphs in Figure 7.

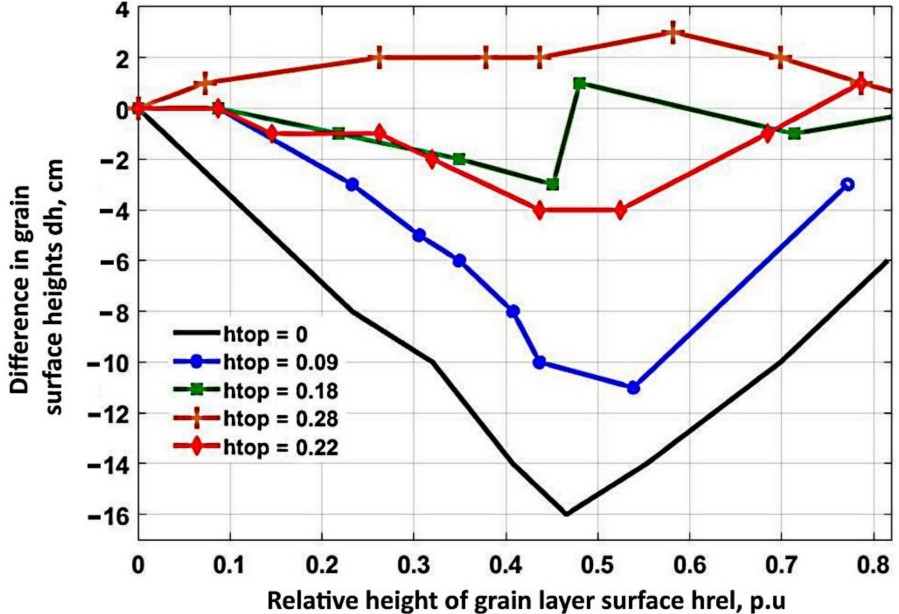

**Figure 7.** Results of the experimental research on the influence of bulk material flow separator length and its positioning on the uneven grain movement in an outlet hopper.

The results of the experimental research (Figure 7) revealed a significant effect of the vertical partition on the uniformity of grain movement in an outlet hopper. The graphs demonstrate that the highest uniformity was obtained for the variants $h_{top} = 0$, $h_{top} = 0.09$ and $h_{top} = 0.28$. The unevenness is oscillatory at $h_{top} = 0.18$ and $h_{top} = 0.22$. In the first half of the hopper height, the height difference is equal to zero or even negative. A negative value of *dh* indicates that grain movement in the right half of the hopper is faster than in the left. In the second vertical half of the hopper, the difference in grain surfaces heights becomes positive, then negative again and finally tends to 0. At $h_{top} = 0.28$, the differences in grain surface heights are always positive, but not more than 4 cm. We focused only on the data for $h_{top} = 0.18$, $h_{top} = 0.22$ and $h_{top} = 0.28$ so as to more precisely define the distances to the upper edge of the hopper, which ensures the minimum uneven grain movement along the hopper's height.

Based on the points obtained from the data for the specified curves, a 3D surface $dh = f(h_{rel}, h_{top})$ was built using the MATLAB application package (Figure 8). Figure 8 shows a fairly complex surface. The data obtained for building the surface are problematic enough to use for deriving a regression equation when finding the best $h_{bot}$ value.

Qualitative and quantitative visual evaluation of such a graph is also difficult. Therefore, for the convenience of the evaluation of the influence of the bulk material flow separator parameters on the uneven grain movement in a hopper, the contour surface of this dependence was also constructed (Figure 9) by means of the Curve Fitting Toolbox graphical interface of the MATLAB application package [30]. Using this package and interpolation methods, the surface shown in Figure 8 was drawn for the obtained experimental data dh, $h_{rel}$, $h_{top}$. The Curve Fitting Toolbox functions allow for interpolation by fitting a curve or surface to the data. Then the resulting surface was modified into a contour plot which is presented in Figure 9.

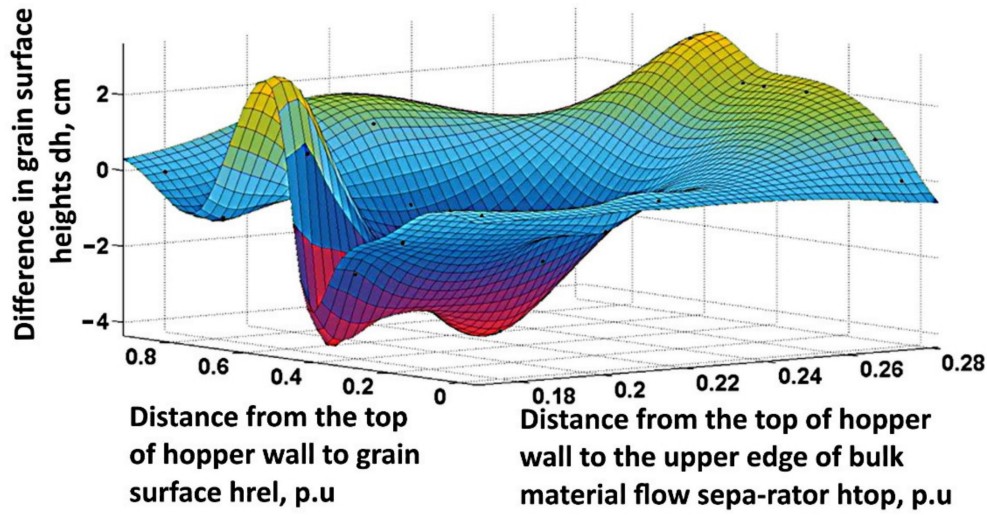

**Figure 8.** Three-dimensional dependency surface dh $= f\left(h_{rel},\ h_{top}\right)$.

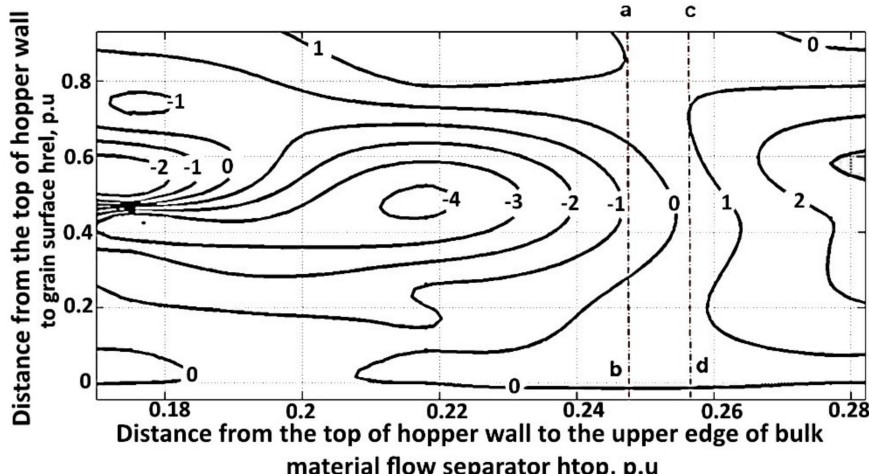

**Figure 9.** Contour surface of dependency dh $= f\left(h_{rel},\ h_{top}\right)$.

Contour surface analysis revealed that the smallest difference in the grain layer surface heights in the vertical halves of the outlet hopper is possible in the range of changes in the distance from the top of the hopper wall to the upper side of the bulk material separator 0.247 . . . 0.256 p.u. This zone is marked as a rectangle abdc in Figure 9.

The resulting interval is quite accurate; however, the expediency of using a value with an accuracy of three decimal places is unknown. The contour plot within the rectangle abdc is considered in order to evaluate the possibility of using simpler sizes. The contour surface is shown in Figure 10.

Figure 10 shows that uneven grain flow along the vertical hopper halves does not exceed $\pm$ 0.5 cm at the distance from the surface outlet hopper, which is 0.247–0.256 p.u. Such an accuracy will ensure the required uniformity of grain movement along the microwave–convective zone. To ensure the uniform movement of grain along the outlet hopper vertical halves, it is assumed for convenience of calculations to install a bulk material flow separator so that its upper side is at a distance of 0.25 of the outlet hopper's total height from the hopper upper edge. This will simplify the calculation of the distance $h_{top}$ and provide fluctuations of the uneven grain movement in an outlet hopper in the range of $\pm$0.5 cm.

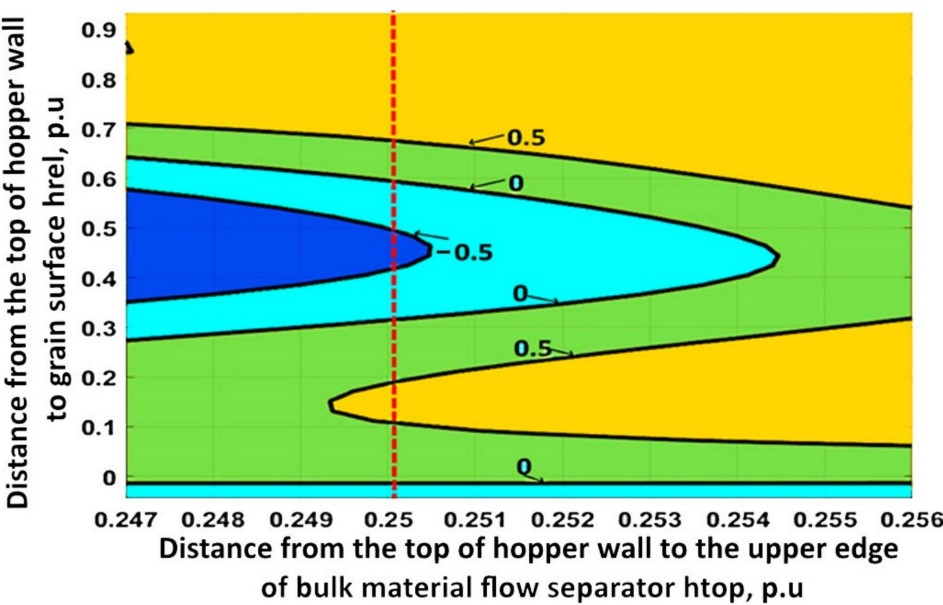

**Figure 10.** Contour surface of the dependence dh $= f(h_{rel}, h_{top})$ in the range of the parameter $h_{top} = [0.247\ 0.256]$.

The conducted experimental studies confirmed the chosen hypothesis that the use of a stationary bulk material flow separator will improve the uniformity of grain movement in an outlet hopper. In this case, the bottom side of the bulk material flow separator should be at a distance of 0.3 of the hopper's total height from the outlet opening, while the upper side should be at a distance of 0.25 of the hopper's total height from the upper edge of the hopper.

The experiment was carried out to assess the reliability of obtained parameters for the bulk material flow separator. A bulk material flow separator was installed in the outlet hopper layout with a height of H = 68 cm. The hopper layout had transparent walls marked into squares 2 × 2 cm in size. The distance from the hopper top wall to the upper side of the separator was 17 cm (0.25 H). The distance from the bulk material flow separator underside to the outlet opening was 20.5 cm (0.3 H). The hopper layout was filled by wheat grain with a moisture content of 14%. Then the outlet valve was opened and the grain flowed out of the outlet hopper. The process of grain movement was filmed with a camera. Thereafter, the resulting record of the process was viewed in slow motion and data on grain movement were recorded.

The efficiency of the use of a bulk material flow separator was also tested on an existing microwave–convective grain processing plant (Figure 11).

Wheat grain was poured into the plant (Figure 12). Along with the opening of the valve of the outlet hopper, the process of grain movement along the height of the microwave–convective zone was recorded with a video camera.

The video was viewed in slow motion and the difference in the distance between the grain surfaces in the right and left vertical parts of the microwave–convective zone was recorded. The obtained data for the hopper model and the existing microwave–convective grain processing plant in the form of graphs are shown in Figure 13.

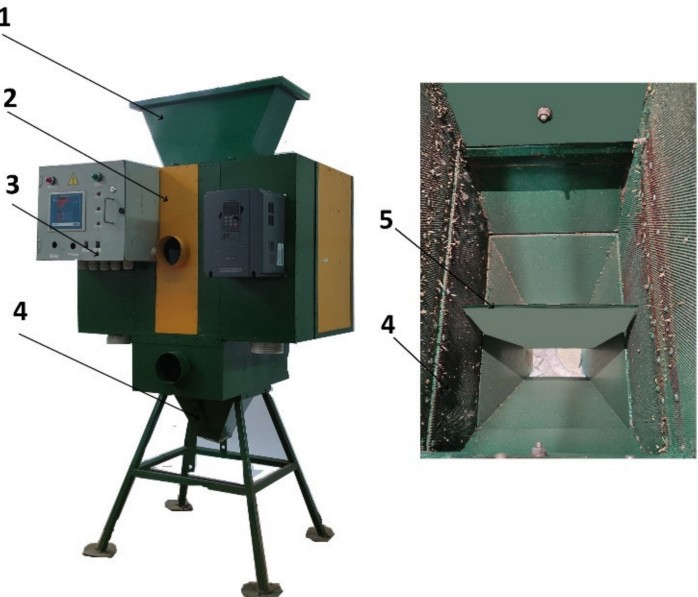

**Figure 11.** External view of a microwave–convective grain processing plant: (1) loading hopper; (2) microwave–convective processing unit; (3) control system unit; (4) unloading hopper; (5) bulk material flow separator.

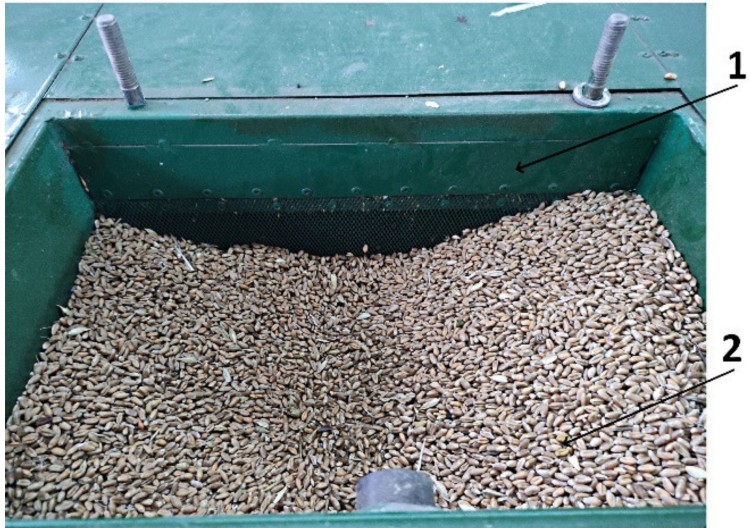

**Figure 12.** Grain in the microwave–convective zone within the experiment: (1) active zone for microwave–convective grain processing; (2) grain in the microwave–convective zone.

The results of experimental studies on the outlet hoppers of both the experimental layout and an existing microwave–convective grain processing plant confirmed the efficiency of using a bulk material flow separator to ensure the uniform movement of grain in the microwave–convective processing zone.

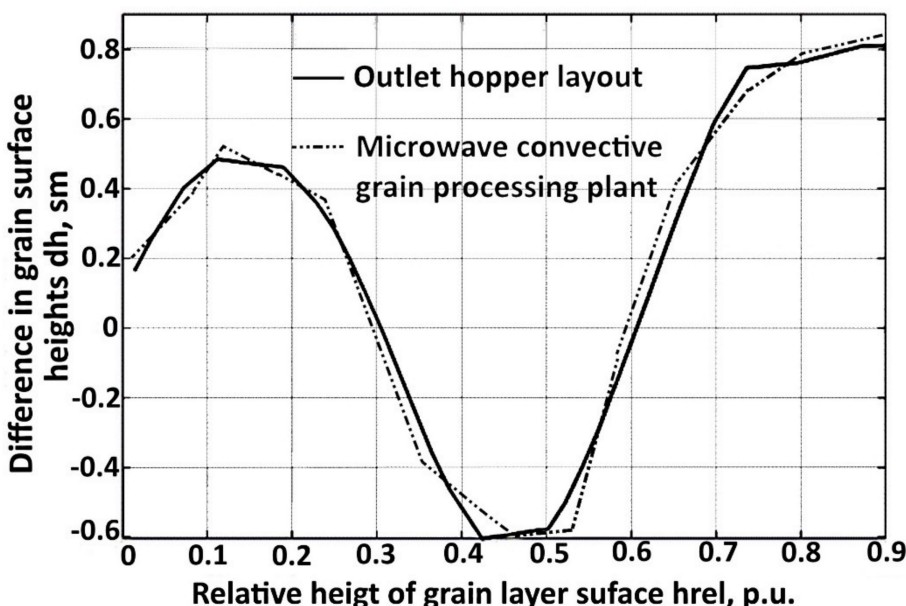

**Figure 13.** Change in uneven heights between the right and left vertical parts of the outlet hopper when moving along the hopper height.

## 5. Conclusions

The results of theoretical studies show that the place of dynamic arch formation and their angles of inclination do not remain constant when grain moves along an outlet hopper. Significant adjustments considering these peculiarities should be made in the elements of the theory for calculating dynamic arches and determining methods for counteracting their formation.

The analysis of hopper dynamic properties using a differential equation describing the change in grain flow from an outlet opening found that most of the parameters included in the equation did not remain constant and depended on the location of the dynamic arch formation in the process of bulk material movement along the hopper's height. Thus, the arch-breaking devices must have a varying vibration frequency of arch-breaking element inside the grain layer when they are used. The effect of the arch-breaking device is local. The device will effectively destroy the dynamic arches only if dynamic arches are formed with the frequency of device operation. At the same time, it is difficult to assume what will be the uniformity of grain movement along outlet hopper vertical halves. Therefore, the use of arch-breaking devices cannot be considered an effective measure to ensure the uniform grain movement in an outlet hopper.

The inclination angle of dynamic arches can be controlled by changes either in the inclination angle of the outlet hopper wall according to grain movement height or in bulk material outflow type. These factors affect the formation frequency of dynamic arches and the inclination angle to the hopper's horizontal axis.

The conducted experimental studies confirmed the chosen hypothesis that the use of a stationary bulk material flow separator changes the type of bulk material outflow and makes it possible to increase the uniformity of grain movement in an outlet hopper. In this case, the bottom side of the bulk material flow separator should be at a distance of 0.3 of the hopper's total height from the outlet opening, while the upper side should be at a distance of 0.25 of the hopper's total height from the upper edge of the hopper.

It was found that the uneven grain movement along outlet hopper vertical halves with the installed bulk material flow separator does not remain constant. It changes as the grain moves from the hopper top wall to the outlet opening. The difference in maximum grain surface heights between the left and right vertical hopper parts is no more than 0.81 cm. The maximum unevenness occurs at the bottom of the outlet hopper. At this moment, only

12 cm of grain remains before the outlet opening of the hopper. Uneven outlet volumes are minimal.

The bulk material flow separator usage can be recommended for use in outlet hoppers of technological machines, for which it is important to observe a uniform movement of bulk material throughout the whole volume of technological space. It is also necessary to take into account that the type of bulk material and its moisture content will affect the rate of grain outflow from an outlet opening. The results obtained in this paper on the use of a bulk material flow separator can be used for all bulk materials if the hopper parameters are calculated taking into account all the properties of the processed material and the methodology ensuring material hydraulic outflow.

**Author Contributions:** Conceptualization, A.N.V.; methodology, D.B.; software, A.A.V.; validation, V.B.; formal analysis, A.A.V., A.N.V. and D.B.; investigation, A.A.V. and D.B.; resources, A.N.V., D.S. (Denis Shilin) and D.S. (Dmitry Shestov); data curation, A.N.V.; writing—original draft preparation, A.A.V.; writing—review and editing, V.B.; visualization, A.A.V. and V.B.; supervision, A.N.V., D.S. (Denis Shilin) and D.S. (Dmitry Shestov); project administration, A.N.V.; funding acquisition, A.N.V., D.S. (Denis Shilin) and D.S. (Dmitry Shestov). All authors have read and agreed to the published version of the manuscript.

**Funding:** This research was supported by the Federal Scientific Agroengineering Center VIM (RF state assignment No FGUN-2022-0004).

**Data Availability Statement:** Not applicable.

**Conflicts of Interest:** The authors declare no conflict of interest.

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
