# Peer review of "Dynamic Arches Destruction by a Bulk Material Flow Separator: A Case Study of the Separator Usage in Microwave Grain Processing Plants"

_agronomy, doi:10.3390/agronomy12050997_

Round 1

Reviewer 1 Report

  1. The study of the bulk material flow movement in the hopper is presented in the article. The hopper can be used in a various types of technological equipment. Therefore, the title of the article needs to be adjusted (it is advisable to remove from the title "microwave grain processing plants").
  2. The article statement ("In addition, when operating the whole variety of this equipment, it is necessary to ensure a uniform supply of bulk material from the hopper to the working bodies. Without this stipulation the productivity of machines decreases (conveyors, grain cleaners, sorting equipment)" (p. 1)) needs clarification. Because, in the case of uneven supply of bulk materials from the hopper to the working bodies of the equipment, the productivity of technological equipment is also unstable, it can increase or decrease.
  3. The article states that "… the formation of arches is due to the compaction of bulk products when they are loaded into the hopper…" (p. 2). It should be noted that the properties of bulk material (including material moisture, material particle size and shape, material fractional composition, material surface, material friction coefficients, etc.) and hopper shape and dimensions have a decisive influence on the formation of bulk material arches in hoppers. The influence of these factors on the formation of bulk material arches in the hopper should be analyzed in more detail in the article.
  4. In the section "2. Materials and Methods" there is no information on the bulk material used in the study. Its physical and mechanical properties, fractional composition and other properties are not given. Also, there is no information on the study methodology and numerical values of the hopper parameters. All information related to the methodology of the study and bulk material should be presented in the section "2. Materials and Methods" of the article.
  5. There is a parameter H in the equation (3) (p. 4). In the explanation to equation (2) it is written that H is the height of the bulk material flow. But in the caption of Figure 1 it is stated that H is outlet hopper height. Also, in the Figures 2 and 3 the outlet hopper height is marked by H. Therefore, it is necessary to make corrections in the article.
  6. It is necessary to check the correctness of the equation (3), because the dimension in the numerator does not coincide with the dimension in the denominator.
  7. The mathematical form of the left side of the equation (?, ??, ??) = ?(â„Ž) is incomprehensible (p. 4). The same applies to the notation of the left side of the equation (?0, ????â„Ž) = ?(â„Ž).
  8. â„Ž is the distance from the top edge of the outlet hopper to dynamic arch (arch formation depth) (p. 4). Since the dynamic arch is asymmetric about the vertical axis of the hopper, it would be appropriate in the article text to indicate where the distance from the top of the hopper to the arch is determined and why this method of distance determining was accepted.
  9. "Equation 5 shows that the hopper is a non-linear object" (p. 5). It needs to be clarified what the authors of the article meant by this statement.
  10. In the equation (6), the entry under the function th needs to be verified, because the dimensions of the numerator and denominator do not match. The same applies to the equation (7), in particular ??? − ? ??. It is also unclear for what reasons the expression ? = ??? − ? ?? was composed for the case ? ≤ ??.
  11. "Data were obtained as a result of processing the video material" (p. 9). It is unclear from the text of the article by which methods the video materials were processed.
  12. The Conclusion should indicate for which bulk materials (with what properties and fractional composition) we can use the proposed separator with experimentally justified its location in the hopper. Because the type of bulk material and its properties can significantly affect the rational parameters and location of the separator in the hopper and, accordingly, the uniformity of the bulk material flow through the hopper.
  13. The first source of references does not contain information about the authors and part of the title of the article:

M. Giyevskiy, V. I. Orobinsky, A. P. Tarasenko, A. V. Chernyshov, and D. O. Kurilov, “Substantiation of basic scheme of grain cleaning machine for preparation of agricultural crops seeds,” IOP Conf. Ser.: Mater. Sci. Eng., vol. 327, p. 042035, doi: 10.1088/1757-899X/327/4/042035.

Author Response

Cover letter to revised version of manuscript “Dynamic arches destruction in microwave grain processing plants using a bulk material flow separator”

At the beginning I would like to thank Reviewers for their time and valuable contributions. All changes that resulted from review were introduced in text are in green. 

-------------------------------------------------------------------------------------
REVIEWER 1
1. The study of the bulk material flow movement in the hopper is presented in the article. The hopper can be used in a various types of technological equipment. Therefore, the title of the article needs to be adjusted (it is advisable to remove from the title "microwave grain processing plants"). 

2. The article statement ("In addition, when operating the whole variety of this equipment, it is necessary to ensure a uniform supply of bulk material from the hopper to the working bodies. Without this stipulation the productivity of machines decreases (conveyors, grain cleaners, sorting equipment)" (p. 1)) needs clarification. Because, in the case of uneven supply of bulk materials from the hopper to the working bodies of the equipment, the productivity of technological equipment is also unstable, it can increase or decrease. 

3. The article states that "… the formation of arches is due to the compaction of bulk products when they are loaded into the hopper…" (p. 2). It should be noted that the properties of bulk material (including material moisture, material particle size and shape, material fractional composition, material surface, material friction coefficients, etc.) and hopper shape and dimensions have a decisive influence on the formation of bulk material arches in hoppers. The
influence of these factors on the formation of bulk material arches in the hopper should be analyzed in more detail in the article. 

4. In the section "2. Materials and Methods" there is no information on the bulk material used in the study. Its physical and mechanical properties, fractional composition and other properties are not given. Also, there is no information on the study methodology and numerical values of the hopper parameters. All information related to the methodology of the study and bulk material should be presented in the section "2. Materials and Methods" of the article. 

5. There is a parameter H in the equation (3) (p. 4). In the explanation to equation (2) it is written that H is the height of the bulk material flow. But in the caption of Figure 1 it is stated that H is outlet hopper height. Also, in the Figures 2 and 3 the outlet hopper height is marked by H. Therefore, it is necessary to make corrections in the article. 

6. It is necessary to check the correctness of the equation (3), because the dimension in the numerator does not coincide with the dimension in the denominator.

7. The mathematical form of the left side of the equation (?, ??, ??) = ?(â„Ž) is incomprehensible (p. 4). The same applies to the notation of the left side of the equation (?0, ????â„Ž) = ?(â„Ž).

8. â„Ž is the distance from the top edge of the outlet hopper to dynamic arch (arch formation depth) (p. 4). Since the dynamic arch is asymmetric about the vertical axis of the hopper, it would be appropriate in the article text to indicate where the distance from the top of the hopper to the arch is determined and why this method of distance determining was accepted.

9. "Equation 5 shows that the hopper is a non-linear object" (p. 5). It needs to be clarified what the authors of the article meant by this statement.

10. In the equation (6), the entry under the function th needs to be verified, because the dimensions of the numerator and denominator do not match. The same applies to the equation (7), in particular ??? − ? –??. It is also unclear for what reasons the expression ? = ??? − ? –?? was composed for the case ? ≤ ??.

11. "Data were obtained as a result of processing the video material" (p. 9). It is unclear from the text of the article by which methods the video materials were processed. 

12. The Conclusion should indicate for which bulk materials (with what properties and fractional composition) we can use the proposed separator with experimentally justified its location in the hopper. Because the type of bulk material and its properties can significantly affect the rational parameters and location of the separator in the hopper and, accordingly, the uniformity of the bulk material flow through the hopper.

13. The first source of references does not contain information about the authors and part of the title of the article: 

M. Giyevskiy, V. I. Orobinsky, A. P. Tarasenko, A. V. Chernyshov, and D. O. Kurilov,
“Substantiation of basic scheme of grain cleaning machine for preparation of agricultural crops seeds,” IOP Conf. Ser.: Mater. Sci. Eng., vol. 327, p. 042035, doi: 10.1088/1757- 899X/327/4/042035. 

---------------------------------------------------------------------------------
RESPOND TO REVIEWER 1. 

Dear Reviewer,
Firstly, we would like to thank You for all remarks. We indicated all issues from review and answers in details for each element. All changes were highlighted in yellow color in manuscript and in track mode.

Q1. The study of the bulk material flow movement in the hopper is presented in the article. The hopper can be used in various types of technological equipment. Therefore, the title of the article needs to be adjusted (it is advisable to remove from the title "microwave grain processing plants").
Response:
The reviewer is absolutely right about the options for using outlet hoppers. The uniformity of bulk material outflow from a hopper is a necessary condition for almost all technological processes where hoppers are used. For this, as a rule, dynamic arch-breaking devices are used. However, for plants using the energy of the microwave field to process bulk material, it is very important to ensure uniform movement of the grain inside the microwave active zone.
We described it in more detail here [A. A. Vasilyev et al., “Effect of Dynamic Bridging on Homogeneous Grain Movement in a Microwave Processing Zone,” Agronomy, vol. 11, no. 10, p. 2014, Oct. 2021, doi: 10.3390/agronomy11102014.]. Therefore, we think that it is important to emphasize the advantages of our research for microwave processing plants. Nevertheless, we consider it quite appropriate to correct the title of the article in the following version:
"Dynamic arches destruction by a bulk material flow separator: A case study of the separator usage in microwave grain processing plants" 

Q2. The article statement ("In addition, when operating the whole variety of this equipment, it is necessary to ensure a uniform supply of bulk material from the hopper to the working bodies. Without this stipulation the productivity of machines decreases (conveyors, grain cleaners, sorting equipment)" (p. 1)) needs clarification. Because, in the case of uneven supply of bulk materials from the hopper to the working bodies of the equipment, the productivity of technological equipment is also unstable, it can increase or decrease. 

Response:
Dear Reviewer, thank you for your comment. We will try to answer your question.
Controlling the performance of machines depending on the volume of material fed to them is not the same as ensuring the required (declared) supply of material to the production line. For example: the capacity of a screen-air cleaner is 5 t/h. As a rule, the speed of the screens is not regulated. A decrease in the supply of grain from the outlet hopper to the screen-air cleaner by 5% (due to the formation of dynamic arches) leads to a loss in productivity of the screen-air
cleaner by 5%.

Q3. The article states that "… the formation of arches is due to the compaction of bulk products when they are loaded into the hopper…" (p. 2). It should be noted that the properties of bulk material (including material moisture, material particle size and shape, material fractional composition, material surface, material friction coefficients, etc.) and hopper shape and dimensions have a
decisive influence on the formation of bulk material arches in hoppers. The influence of these factors on the formation of bulk material arches in the hopper should be analyzed in more detail in the article.
Response:
Dear Reviewer, you are absolutely right when you speak about the influence of all the factors he has listed on the formation of arches. We described this issue in sufficient detail in a previous article [A. A. Vasilyev et al., “Effect of Dynamic Bridging on Homogeneous Grain Movement in a Microwave Processing Zone,” Agronomy, vol. 11, no. 10, p. 2014, Oct. 2021, doi: 10.3390/agronomy11102014.], so it is why we omitted this in the proposed manuscript. Nevertheless, we consider it quite appropriate to take into account your comment.

Added (lines 49-51)
"In turn, the formation of arches is due to the compaction of bulk products when they are loaded into the hopper depends on the seed culture, their granulometric composition, and humidity [12]”

Q4. In the section "2. Materials and Methods" there is no information on the bulk material used in the study. Its physical and mechanical properties, fractional composition and other properties are not given. Also, there is no information on the study methodology and numerical values of the hopper parameters. All information related to the methodology of the study and bulk material
should be presented in the section "2. Materials and Methods" of the article.
Response:
Dear Reviewer, thank you for your crucial concern. We agree that we had missed this information in the presented version of paper. We added it in the revised manuscript. 

Added (lines 108-130)
The experimental model of outlet hopper was made in such a way as to be able to visually observe and record the grain movement along the hopper heigh during its unloading. To do this, the experimental model (layout) was made of plexiglass. The side walls of the hopper layout were divided into squares 2x2 cm in size so that it was possible to accurately determine the coordinates of grain movement. 

The experimental model was a part (section) of the outlet hopper 5 cm thick. The parameters of the model were calculated in accordance with the parameters of technological plants for microwave-convective grain processing (its length and width). The productivity of the plant for microwave-convective drying of spiked crops was 15 t/h. The inclination angle of bunker walls was calculated based on the fact that the plant would process the grain of wheat, barley, sunflower [12]. For these crops, the following were taken into account: 

Laying angle was 17ï‚° for wheat, 18ï‚° for barley, 16ï‚° for sunflower; Internal friction angle was 16ï‚° for wheat, 15ï‚° for barley, 19ï‚° for sunflower; Reduced friction angle was 27ï‚° for wheat, 39ï‚° for barley, 26ï‚° for sunflower. As a result of calculations, the width of outlet opening was adopted equal to 90 mm while the
height of the hopper was to 690 mm. A cut was made along the vertical central axis of the hopper, which made it possible to build a plate inside so as to divide the layout into two equal parts. The total volume of the bunker was 0.0082 m3.

The transparent walls of the experimental model allowed filming the movement of grain along the hopper height. The video played in slow motion in order to record the change in the height of the moving grain layer. Subsequently, the obtained data were presented in the form of graphs. 

Q5. There is a parameter H in the equation (3) (p. 4). In the explanation to equation (2) it is written that H is the height of the bulk material flow. But in the caption of Figure 1 it is stated that H is outlet hopper height. Also, in the Figures 2 and 3 the outlet hopper height is marked by H. Therefore, it is necessary to make corrections in the article.

Q6. It is necessary to check the correctness of the equation (3), because the dimension in the numerator does not coincide with the dimension in the denominator.

Q10. In the equation (6), the entry under the function th needs to be verified, because the dimensions of the numerator and denominator do not match. The same applies to the equation (7), in particular ??? − ? –??. It is also unclear for what reasons the expression ? = ??? − ? –?? was composed for the case ? ≤ ??.

Response to Q5, Q6 and Q10:
Dear Review, thank you for your meticulous reviewing. We made typos by accident which fortunately did not affect the correctness of calculation. The mistakes in equation (3), (6), (7) were corrected

Corrected (lines 257, 301, 308) 

Added (lines 311-317):
The system of equations (7) reflects the change in the flow rate of bulk material at the outlet of hopper in two stages. The first stage represents the transient process, when the outlet valve has changed its position. As a result, the flow rate of bulk material changes to the value steady state value of the bulk material flow rate after the completion of the transition process ???. The second stage is the grain outflow from the outlet opening, when the valve remains stationary while the flow rate changes only due to the formation of dynamic arches in a hopper.

Q7. The mathematical form of the left side of the equation (?, ??, ??) = ?(â„Ž) is incomprehensible (p. 4). The same applies to the notation of the left side of the equation (?0, ????â„Ž) = ?(â„Ž).
Response:
Dear Review, thank you! We meant the following: ? = ?(â„Ž), ?? = ?(â„Ž), ?? = ?(â„Ž), and ?0 = ?(â„Ž),????â„Ž = ?(â„Ž). Changes applied.
Corrected (lines 264, 279) 

Q8. â„Ž is the distance from the top edge of the outlet hopper to dynamic arch (arch formation depth) (p. 4). Since the dynamic arch is asymmetric about the vertical axis of the hopper, it would be appropriate in the article text to indicate where the distance from the top of the hopper to the arch is determined and why this method of distance determining was accepted.
Response:
Dear Reviewer, thank you for your concern. The coordinates of the points of the surface of the dynamic arches are determined in the following sequence [17]: Set the distance from the upper edge of the hopper h; Set the coordinate along the hopper width; Calculate the coordinate by the hopper height. Thus, a set of points is obtained for one distance h. At each of these points, the values of h will be equal. In figure 4, the distance h is shown arbitrarily, only to indicate that the arches refer to different values of h.

Q9. "Equation 5 shows that the hopper is a non-linear object" (p. 5). It needs to be clarified what the authors of the article meant by this statement.

Response:
Dear Reviewer, thank you for your concern. The clarification of the statement is that nonlinear object are objects whose dynamic properties are described by non-linear differential equations. This is exactly what equation (5) is.

Q11. "Data were obtained as a result of processing the video material" (p. 9). It is unclear from the text of the article by which methods the video materials were processed.
Response:
Dear Reviewer, we have made appropriate explanations in the section "Materials and Methods" for the revised manuscript.
Added (lines 127-130):
“The transparent walls of the experimental model allowed filming the movement of grain along the hopper height. The video played in slow motion in order to record the change in the height of the moved grain layer. Subsequently, the obtained data were presented in the form of graphs.”

Q12. The Conclusion should indicate for which bulk materials (with what properties and fractional composition) we can use the proposed separator with experimentally justified its location in the hopper. Because the type of bulk material and its properties can significantly affect the rational parameters and location of the separator in the hopper and, accordingly, the uniformity of the
bulk material flow through the hopper.
Response:
Dear reviewer, we tried to take into account your comment in the conclusion.
Added (lines 679-684):
It is also necessary to take into account that the type of bulk material and its moisture content will affect the rate of grain outflow from an outlet opening. The results obtained in this paper on the use of a bulk material flow separator can be used for all bulk materials if the hopper parameters are calculated taking into account all the properties of the processed material and the methodology ensuring material hydraulic outflow. 

Q13. The first source of references does not contain information about the authors and part of the title of the article:
M. Giyevskiy, V. I. Orobinsky, A. P. Tarasenko, A. V. Chernyshov, and D. O. Kurilov,
“Substantiation of basic scheme of grain cleaning machine for preparation of agricultural crops seeds,” IOP Conf. Ser.: Mater. Sci. Eng., vol. 327, p. 042035, doi: 10.1088/1757- 899X/327/4/042035.
Response: 

Dear review, thank you for your comment. We do not know why the information on this
reference was lost when preparing the PDF version of manuscript under your consideration. In the revised version, the problem is ameliorated.

Reviewer 2 Report

The paper presents a theoretical study of the crucifixion of material from the hopper.
For a magazine called Agronomy, it is necessary to confirm theoretical study with a practical study. Therefore, it is necessary to perform research in real conditions, or at least to perform a more detailed simulation of this research than the one briefly presented in the last paragraph of the chapter 4. Experimental Research (line 333-348).
With this added practical research, paper can be accept.

Author Response

Dear Reviewer,

Firstly, we would like to thank You for remarks. We answered your concern in details. Please, see the attached file.

All changes in manuscript were highlighted in yellow color and in track mode. 

Round 2

Reviewer 2 Report

Manuscript has been sufficiently improved and it can be accept in present form.